# Relationships between Heavy Metal Concentrations in Greater Celandine (*Chelidonium majus* L.) Tissues and Soil in Urban Parks

**DOI:** 10.3390/ijerph20053887

**Published:** 2023-02-22

**Authors:** Oimahmad Rahmonov, Dorota Środek, Sławomir Pytel, Natalina Makieieva, Teobald Kupka

**Affiliations:** 1Institute of Earth Sciences, Faculty of Natural Sciences, University of Silesia, Będzińska 60, 41-200 Sosnowiec, Poland; 2Institute of Social and Economic Geography and Spatial Management, Faculty of Natural Sciences, University of Silesia in Katowice, Będzińska 60, 41-200 Sosnowiec, Poland; 3Faculty of Chemistry, University of Opole, Oleska 48, 45-052 Opole, Poland

**Keywords:** medicinal plant, soil properties, soil–vegetation link, urban park, soil contamination

## Abstract

Anthropogenic ecological ecosystems create favourable conditions for the growth of the nitrophilous medicinal species *Chelidonium majus* in six urban parks in Southern Poland. This study focuses on the concentrations of trace elements in the soils, leaves, stems, and rhizomes of greater celandine. The soil samples were taken only in the humus horizon (A), which averaged approximately 15 cm in thickness under the clumps of *Ch. majus*. Regarding the reaction, the soil samples tested can be described as slightly acidic (5.6–6.8 in KCl) to alkaline (7.1–7.4 in H_2_O). Organic carbon content at all sites is high, ranging from 3.2% to 13.6%, while the highest total nitrogen (Nt) content is 0.664%. The average total phosphorus (Pt) content in all samples is 548.8 mg/kg (and its range is 298–940 mg/kg), such values indicating its anthropogenic origin. In terms of heavy metals, Zn has the highest content in the analysed soil samples compared to the other elements, and its range is from 394.50 mg/kg to 1363.80 mg/kg in soil. In rhizomes, Zn also has the highest values (178.7–408.3 mg/kg), whereas, in stems and leaves, it varies (from 80.6 to 227.5 and from 57.8 to 297.4 mg/kg, respectively). Spearman’s rank correlation showed high correlations between the content of Pb, Zn, Cd, and As in the soil and rhizomes of *Ch. majus*. Despite soil contamination with Pb, Cd, and Zn, *Ch. majus* does not accumulate them in its tissues. However, the translocation of Hg and Cr from rhizomes to leaves was observed. The different concentrations of metals in each park result from the degree of diversity of the parent rocks on which the soil was formed.

## 1. Introduction

*Chelidonium majus* L., also known as greater celandine, is a plant species belonging to the *Papaveraceae* family. *Ch. majus* has many common names, such as celandine, greater celandine, celandine poppy, killwort, rock poppy, swallow-wort, and tetter-wort [1]. The name Chelidon (χελιδον) in Greek means swallow, and, with this bird, greater celandine was associated. Ancient writers noted that the celandine flower bloomed when the swallows returned and faded when they left [2].

This botanical family is rich in specific alkaloids, some of which are important in medicine. A characteristic feature of this family is the presence of the mentioned alkaloids in the orange latex that outflows when fresh stems are cut. During drying, the plant loses these alkaloids.

In nature, *Ch. majus* grows in Asia, Europe, and the Mediterranean [3]. It is also distributed in North America [3], where it was introduced in 1672 by the colonialists as a medicine for warts. Greater celandine was naturalised worldwide in the temperate zone and often occupies an anthropogenic habitat with high nitrogen content; therefore, it is called a nitrophilous species.

The species has a long tradition of being used mainly in folk medicine. It has been used in herbal medicine since Dioscorides and Pliny the Elder’s time in the 1st century AD [4,5]. *Ch. majus* is a herbal plant highly praised for its therapeutic potential in western phytotherapy, traditional Chinese medicine [1], and European countries [6]. In various complementary and alternative medicine systems, including homeopathy, different parts of this plant are used to treat gastric ulcers, gastric cancer, oral infection, liver diseases, general pain, and various skin disorders [7,8]. Moreover, in traditional medicine, *Ch. majus* has been well known for many centuries. It has been used for a long time in hepatobiliary disorders: gall bladder and digestive dysfunctions, dyspeptic complaints, and spasms in phytotherapy [1,9].

The plant contains, as major constituents, isoquinoline alkaloids (such as sanguinarine, chelidonine, chelerythrine, berberine, protopine, and coptisine), flavonoids, and phenolic acids [10,11,12]. Other compounds structurally unrelated to the alkaloids have been isolated from the aerial parts: several flavonoids and phenolic acids. *Ch. majus* extracts and its purified compounds exhibit interesting antiviral, antitumour, and antimicrobial properties [10,13]. Both crude extracts of *Ch. majus* and the purified compounds derived from it exhibit a wide variety of biological activities (anti-inflammatory, antimicrobial, immunomodulatory, antitumoral, choleretic, hepatoprotective, analgesic, etc.), which are in concordance with the traditional uses of *Ch. majus*. This species exhibits multiple biological actions, such as antiviral, antitumour, antibacterial, antifungal, antiprotozoal, anti-inflammatory, and antispasmodic effects [5,14,15,16]. Extensive phytochemical investigation revealed the presence of a variety of alkaloids in different parts of *Ch. majus* [1]. More than 70 compounds have been isolated and identified from this species, including alkaloids, flavonoids, saponins, vitamins (e.g., vitamins A and C), mineral elements, sterols, acids, and their derivatives [17].

The use of the therapeutic plant in various fields of medicine is well known. The beneficial properties of plants may be due to their organic agents and inorganic mineral elements. Measurement of the trace and major element content in plant drugs may be relevant in view of human health, animal health, environmental relations, etc. This fact has great significance since approximately half of the plant drugs available in the market originate from natural habitats [18,19]. The element content of herbs may refer to soil pollution, the soil type on which the plant grows, or air pollution.

The organic compounds and main bioactive agents of plants are generally known, but the element composition and concentrations of elements are unknown in most cases. The determination of microelement content is important in view of plant, animal, and human health, and environmental aspects as well. Therefore, the measurement of toxic, essential, and non-essential elements in plant drugs may be significant in environmental, toxicological, and phytotherapeutical aspects [20,21,22,23].

The studies on *Ch. majus* were mainly connected with its medicinal properties and effects in use based on the alkaloids that the species biosynthesise [1]. In addition to organic compounds, the dissoluble mineral elements of herbs may also have a role in therapy. A survey of selected metals was performed directly from dried plant tissues and various extracts derived from *Ch. majus* [24,25,26,27,28]. Some research was conducted on the toxic effects of heavy metals on the germination and growth of *Ch. majus* seedlings in urban areas [29].

To date, there have been no studies comparing the toxic potential metal content of *Ch. majus* tissues and in the soil in which the plant grows. In general, the habitats of *Ch. majus* are slopes, forest edges, meadows, roadsides, stone crevices, shady banks, stream banks, and urbanised areas, which are not devoid of various pollutants [30]. It is a typical ruderal plant that grows in habitats with high concentrations of nitrogen and phosphorus around locations transformed by human impacts.

We assume that a plant growing in anthropogenic soils containing heavy metals may accumulate them in its tissues. Accumulating toxic metals in medicinal plants can indirectly affect human health and environmental quality. Therefore, the aim of the present study is to determine the chemical composition of the leaves, stems, and rhizomes of *Chelidonium majus* (a) and the soil (b) in which they grow for heavy metals.

## 2. Materials and Methods

### 2.1. Study Area

During the field work, six locations with soil degradation resulting from human activity were selected for further research. The research was carried out in an allotment (APrzem-1) and urban parks (PSchoe-2, PZiel-3, PLes-4,5, PSiel-6) in Southern Poland, as outlined in Table 1. These parks differ in terms of their degree of disturbance, which is related, in part, to their place of origin.

All sampling sites belong to the Upper Silesian Industrial Region’s central part. We selected five sites to be analysed within the city of Sosnowiec, comprising four sampling points in different parks (PSchoe-2—Schöen Park, PLes-4,5—Leśna Park, PSiel-6—Sielec Park) and one point in allotment gardens near Przemsza river (APrzem-1). One site was selected in Dąbrowa Górnicza, a town adjacent to Sosnowiec. Samples were taken there from Zielona Park (PZiel-3). The whole study area has a diversified topography, which was transformed as a result of anthropogenic activity—the examined area lies within the range of a few coal mining areas. The parent rocks for the soil developing in the study parks are in part materials transported from external sources as reclamation material. Natural bedrock is only present in the Zielona Park.

All sampling sites are located in the industrial impact zone (Figure 1) and are intensively used by the region’s population for sports and recreational purposes. Due to high industrialisation, the vegetation cover has been significantly transformed in all studied sites [31]. Analyses of heavy metal concentrations in the soil and tissues of *Ch. majus* were conducted at all of these sites.

### 2.2. Morphological Traits of Ch. majus

Herb, perennial plant, 30–60 (−100) cm tall and at the breakdown of the shoot, producing drops of thick, milky juice that immediately becomes orange–red in the air. The whole plant contains yellow to orange latex. Taproot conical, stout, and many lateral roots. Stems cymose, branched, and ribbed. Basal leaves few, caducous; petiole 2–5 cm, pubescent or glabrous, base ampliate to sheath; blade glaucous abaxially, green adaxially, obovate–oblong or broadly obovate, 8–20 cm, abaxially sparsely shortly pubescent, adaxially glabrous, pinnatisect; lobes 2–4 pairs, obovate–oblong, irregularly parted or lobed; lobe margin crenate [3]. Inflorescences: peduncle 2–10 cm. Flowers: pedicels 5–35 mm; sepals to 1 cm; petals bright yellow, obovate to oblong, to 2 cm wide. Seeds dark brown, shiny, ovoid, ca. 1 mm or shorter, alveolate. The flowers appear from early spring to late summer, from May/April to September.

### 2.3. Plant and Soil Sampling

Samples for chemical analyses were taken from the rhizome zone of *Ch. majus* by shaking the soil from the plant rhizomes. This zone represents the humus horizon (A), and its thickness averaged 20 cm. In the laboratory, air-dried samples were sieved and analysed, following the standard procedures given by Bednarek et al. [32]. The pH values were measured potentiometrically in H_2_O and in 1N KCl using a glass electrode. The following measurements and methods were carried out: total organic carbon (Corg.) according to Tiurin’s method; total Nt content using the Kjeldahl method; total phosphorus (Pt) by Bleck’s method as modified by Gebhardt [32]; available phosphorus (Pavail) by Egner–Riehm’s method; and available potassium (Kavail) and magnesium (Mgavail) according to the PN-R-04023/23 norm.

The granulometric composition of the samples was determined using standard grain size analysis with a fixed mesh size sieve column. The test was carried out with a set of sieves with different mesh sizes: 20 mm, 10 mm, 5 mm, 2 mm, 1 mm, 0.5 mm, 0.25 mm, 0.1 mm, 0.05 mm, 0.02 mm, 0.006 mm, and 0.002 mm. The mass of the sample remaining in each sieve was calculated as the percentage of grains of a given size in the total mass of the sample [32].

The leaves, stems, and rhizomes of *Ch. majus* were sampled at the end of the vegetation season in late September and early October. At each study site, the material was collected from 5 individuals. Later, the material was mixed to form 1 sample, which was further investigated. The preliminary preparation of the samples for analyses involved washing the plant material with distilled water (the use of stronger agents can remove heavy metals), drying at room temperature for two weeks and at 105 °C for 4 h, followed by homogenisation. Sampling and preparation procedures followed the instructions given by MacNaeidhe [33] and Markert [34].

The total composition of heavy elements lead (Pb), cadmium (Cd), zinc (Zn), iron (Fe), manganese (Mn), chromium (Cr), copper (Cu), nickel (Ni), arsenic (As), and mercury (Hg) in plant material and soil was measured using inductively coupled plasma optical emission spectrometry (ICP-OES) after wet mineralisation in nitrohydrochloric acid (3HCl + HNO_3_). The analyses were performed in the ACME Laboratory (Vancouver, Canada) using the AQ250_EXT (soils) and VG105_EXT (plant tissues) procedures and 5 g samples. All plant tissue and soil samples were analysed in triplicate for all the investigated parameters, and mean values were calculated.

Bioaccumulation factors (BAFs) for *Ch. majus* parts (rhizomes, stems, leaves) were calculated using the following formula: BAF = C_b_/C_n_, where C_b_ and C_n_ are the concentrations of metals in plant parts and soil, respectively [35]. For this purpose, the concentrations of metals from the humus horizons (i.e., the rhizomes zone) of the analysed soils were considered.

The translocation factor (TF) was obtained using the equation TF = C_n_/R_n_, where C_n_ is the element content in the aboveground parts of the plant, and B_n_ is the concentration of the same element in the roots [36,37]. The results can be divided into four classes: low contamination factor (<1), moderate contamination factor (1–3), considerable contamination factor (3–6), and very high contamination factor (>6).

### 2.4. Statistical Analyses

Spearman’s rank correlation coefficient was applied to check whether there was any relationship between the concentrations of the selected elements in the tissues (rhizome, stem, and leaves) of *Ch. majus* and the soil materials in the study surfaces. The exact values of the correlation coefficient were calculated for alpha = 0.05. The Spearman rank correlation coefficient is used to analyse the interdependence of objects in terms of a two-dimensional trait (X, Y). Assuming that we are examining n objects described by two characteristics, these objects must be ordered with regard to the values of each characteristic separately. All statistical analyses were performed using Statistica 14.0.0.15 software.

## 3. Results

### 3.1. Soil Physico-Chemical Features

The granulometric composition varies between the study sites (Table 2). The dominant fractions at all sites are coarse sand (0.5 < d ≤ 1.0), medium sand (0.25 < d ≤ 0.5), and (0.25 < d ≤ 0.5) and fine sand. The significant proportion of very fine sand (0.05 < d ≤ 0.1) favours the soil structure. The highest proportions of rock fragments and gravel fractions were found in the PSchoe-2 (14.9% and 8.3%, respectively) and PLes-4 (16.2% and 9.0%, respectively) samples.

Regarding the reaction, the soil samples tested can be described as slightly acidic (5.6–6.8 in KCl) to alkaline (7.1–7.4 in H_2_O). Only a slight acid reaction in water was found from sites PZiel-3 (pH 6.4) and PSiel-6 (pH 6.3). This type of reaction influences the values of hydrolytic acidity (H_h_) and the content of acid cations (Table 3). At slightly acidic reactions, H_h_ obtains higher values. Such regularities were found at site PSiel-6, where the pH (6.3) and hydrolytic acidity values were 4.72 cmol (+)/kg. A similar situation was found at site PZiel-3.

The results of loss on ignition indicate a high proportion of organic matter in the upper layers of the examined anthropogenic organic horizon. Its highest values were found at the foot of the artificial embankment at sites PSiel-6 (26.56%) and PLas-4 (20.27%), where abundant *Ch. majus* grew. High values of this indicator were also observed at the other sites (Table 3). In all samples, the organic carbon content is high, ranging from 3.2% (PLes-4) to 13.6% in the PSiel-6 sample. Total nitrogen (Nt) has varied content at the studied sites. The highest levels of Nt were recorded at Sielec Park (0.664%) and Leśna Park (0.601%). Despite high ignition loss values (11.83%) at this site (PSchoe-2), nitrogen had the lowest value (0.223%). Carbon to nitrogen ratios ranged from low (5–20, narrow) to high (37–40, wide) values. Significant amounts of total nitrogen (Nt) accumulate in all analysed organs. They are as follows: rhizomes (2.783%), stems (2.469), and leaves (2.296), with an average of 2.516%, also indicative of the high nitrogen content of the soil.

The results obtained from analysing the available content of Mg, K, and P in the studied sites are presented in Table 3. The average content of magnesium was 235.1 mg/kg (139.0–329.5 mg/kg), potassium 234.5 mg/kg (178–305 mg/kg), and phosphorus 88.3 mg/kg (39.0–146.6 mg/kg). The highest content of Mg (329 mg/kg), K (305 mg/kg), and assimilable phosphorus (146.6 mg/kg) was found at sites located at Leśna Park (PLes-4 and PLes-5). The highest total phosphorus content (Pt 940 mg/kg) was found in sample PSiel-6, whereas the lowest was in sample PZiel-3. Its average content in all samples was 548.8 mg/kg. Such values indicate its anthropogenic origin.

Based on Spearman’s rank correlation coefficient, the study plots showed very strong positive correlations between loss on ignition and Pavail and Pt (r = 0.771, *p* = 0.05) due to the high content of organic matter of anthropogenic origin. On the other hand, Nt content showed a high positive correlation (r = 0.886, *p* = 0.05) with exchangeable acidity (H_h_) and hydrogen ion content (H^+^). A strict high correlation was also found for Mgavail-Kavail (r = 0.929, *p* = 0.05) and Pavail-K (r = 0.829, *p* = 0.05). Such high correlative relationships are due to anthropogenic soil enrichment from various sources, mainly due to the maintenance and fertilisation of ornamental plants in the park area.

A high negative correlation was found between the following variables: KCl-Nt (r = −0.657, *p* = 0.05) and KCl-H_h_ (r = −0.886, *p* = 0.05). This is due to the high soil reaction and the presence of limestone crumbs and concretions of anthropogenic origin. Other correlations are shown in Appendix A.

### 3.2. Content of Heavy Metals in Soil

The content of potentially toxic metals in the surface soil layer and in the analysed tissues (rhizomes, stems, and leaves) of *Ch. majus* varied at all sites. Zn had the highest content in the analysed samples compared to other elements (Table 4). It ranged from 394.50 mg/kg (PZiel-4) to 1363.80 mg/kg (PLes-4), and its (Zn) mean value was 922.13 mg/kg for all soil samples tested. The highest Pb content was found at sites APrzem-1 (526.55 mg/kg), PSchoe-2 (373.57 mg/kg), and PLes-4 (346.23 mg/kg), and the lowest occurred at PZiel-3 (119.00 mg/kg). The lowest Pb, Zn, and Cd values occurred at the hornbeam forest’s edge (Table 1 and Table 4).

Cd values range from 3.13 mg/kg (PZiel-3) to 13.46 mg/kg (PLes-4). Cadmium, zinc, and lead exceed the acceptable standards for recreational areas and cultivated soils given by the regulations of the Ministry of the Environment on 5 September 2016 regarding the assessment method of the pollution of the Earth’s surface (Journal of Laws, 2016, item 1395). Mn content ranged from 344.00 mg/kg (PZiel-3) to 716.00 mg/kg at the APrzem-1 site. The lowest content of Cu (21.90 mg/kg), Ni (9.40 mg/kg), As (8.50 mg/kg), and Hg (112 ppb/0.112 mg/kg) was found at Zielona Park (PZiel-3), and the overall range at all sites was as follows: 21.90–155.79 (Cu), 9.40–33.80 (Ni), 8.50–23.10 mg/kg (As), and 112–330 ppb (Hg). Cu exceeds the limits almost at all examined sites (except PZiel-3) (Table 4), and its permissible concentration is 26 mg/kg. Chromium occurs within acceptable limits and is not shown as a soil contaminant in the range of 13.50–25.00 mg/kg (Table 4). The average Fe content in the epipedons of the studied parks is around 1.99%. The highest amounts are found in Schöen Park (2.76%) and the lowest in Leśna Park (1.07%). It is an essential element for soil-forming processes and significantly influences the soil properties in terms of chemistry and soil morphology.

### 3.3. Content of Heavy Metals in Ch. majus Tissues

The content of the analysed tissues in terms of potentially toxic metals is shown in Table 5. The most metals accumulated in the rhizomes and leaves of the studied plants. Pb values in the rhizome ranged from 8.36 mg/kg (PZiel-3) to 33.92 mg/kg (APrzem-1), while, in the leaves, the content ranged from 4.99 to 20.60 mg/kg (PZiel-3 and PLes-4, respectively). In the stem, Pb occurs almost two times less.

Cd and Zn also have the highest values in the rhizomes (3.30 and 408.30 mg/kg, respectively), while, in the leaves, Cd reaches lower content (max. 1.07 mg/kg). Zn values are mostly more elevated in the stems (PSiel-6—158.90 mg/kg, APrzem-1—150.8 mg/kg, PZiel-3—91.10 mg/kg, and PLes-5—80.60 mg/kg) than in the leaves (Table 5).

Mn values range from 45 to 181 mg/kg (rhizomes), 20 to 72 mg/kg (stems), and in leaves from 42 to 293 mg/kg. Higher Cu content is found correspondingly in rhizomes (range: 7.46–19.53 mg/kg) than in leaves (7.36–14.24 mg/kg) and stems (3.30–5.80 mg/kg). Ni content is highest in rhizomes and ranges from 1.30 to 3.10 mg/kg, except in one case, where higher values were found in leaves (PSchoe-2—8.40 mg/kg). The lowest content of Ni was accumulated in the stems (0.30–0.90 mg/kg). The range of Cr varied in the analysed tissues, with the highest content mostly recorded in leaves (from 1.90 to 16.10 mg/kg) than in rhizomes (2.10–4.00 mg/kg) (Table 3). The range of its content in stems varied from 1.50 to 2.60 mg/kg. Pb, Cd, Zn, Mn, Fe, Cu, Ni, and Cr content exceed the limits permitted for food and medicinal plants according to the FAO/WHO Guidance Note.

Higher mercury values were found in the leaves, ranging between 27 and 54 ppb (Table 5). The opposite situation was observed for As. The range of As occurrence was as follows: rhizomes (0.60–2.60 mg/kg), stems (0.00–0.60 mg/kg), and leaves (0.20–1.50 mg/kg).

The average iron content in rhizomes was 0.12% (1200 mg/kg), in stems 0.05% (500 mg/kg), and in leaves 0.1% (1000 mg/kg).

### 3.4. The Correlation of Metal Content in the Soil–Plant System

Spearman’s rank correlation showed, for the analysis of metal content in soil and rhizomes (*Ch. majus*), high correlations between the following elements: Pb-As (r = 0.829, *p* = 0.05), Pb-Cd (r = 0.943, *p* = 0.05), Cd-As (r = 0.886, *p* = 0.05), Pb-Zn (r = 0.943, *p* = 0.05), Cu-Hg (r = 0.889, *p* = 0. 05), Cr-Ni (r = 0.886, *p* = 0.05), As-Cr (r = 0.714, *p* = 0.05), comparing the content of the first element in the soil and the second in the rhizomes. Such high correlations are due to the high content of heavy metals in the anthropogenic soils, providing a habitat for the species under study. In general, high correlations were found in the soil–rhizome system in all analysed sites (Appendix A).

In contrast to the rhizomes, the relationship between the elemental content of the soil and the stem is slightly lower (Appendix A). This is due, among other things, to the seasonal dying of the aerial parts of the plants. The strongest positive correlation was found for Pb-Cd (r = 0.771, *p* = 0.05), Pb-Zn (r = 0.657, *p* = 0.05), Pb-Hg (r = 0.754, *p* = 0.05), Zn-Cd (r = 0.771, *p* = 0.05), Zn-Cu (r = 0.600, *p* = 0.05), Mn-Cd (r = 0.714, *p* = 0.05) Ni-Cd (r = 0.943, *p* = 0.01), As-Cd (r = 0.886, *p* = 0.05), As-Zn (r = 0.771, *p* = 0.05). A low correlation was found for Fe-Cr (r = 0.029, *p* = 0.05), Hg-Cr (r = 0.314, *p* = 0.05), and Cd-Cr (r = 0.257, *p* = 0.05) in the soil–stem system (Appendix A). For As in the stem, a negative correlation ranging from r = −0.093, *p* = 0.05 (As-stem, Cr-soil) to r = −0.833, *p* = 0.05 (As-stem, Hg-soil, Appendix A) was found.

Based on Spearman’s rank correlation, a high correlation was found between soil Cu and other elements, such as Hg in leaves (r = 0.812, *p* = 0.05) and Zn in leaves (r = 0.771, *p* = 0.05). High correlations were found for Cd in leaves with the following elements in soil: Ni (r = 0.943, *p* = 0.01), As (r = 0.886, *p* = 0.05), Pb (r = 0.771, *p* = 0.05), Zn soil (r = 0.771, *p* = 0.05). Other correlations in the soil material–leaf system of *Ch. majus* are presented in Appendix A.

### 3.5. Bioaccumulation Factor

According to the ranges for bioaccumulation factors (Table 6) given by Sekabira et al. [35], greater celandine should be considered a low accumulator of Pb, As, and Fe. For the other heavy metals (Cd, Zn, Mn, Cu, Cr, Hg), *Ch. majus* is a moderate accumulator. The highest BAF value in greater celandine leaves was recorded for Cr and Hg, whereas, for rhizomes, this index was highest for Zn and Cd. The stems had the lowest bioaccumulation factors of the tested plant tissues. The only factor for Zn in stems exceeds the threshold set between low and moderate accumulators.

### 3.6. Translocation Factor

The calculated translocation factors (Table 7) show that heavy metals are not translocated from rhizomes to stems. However, one can observe the translocation (TF > 1) of the selected heavy metals to the leaves. The most translocated heavy metals are Hg and Cr. Increased translocation factors were also noted for Mn and Ni.

## 4. Discussion

The formation of urban soils was conditioned in the initial development period by the action of natural soil-forming factors, the intensity of which decreased over time in favour of human activity. This led to the loss or alteration of some of the morphological characteristics and bio-physicochemical properties of soil created by natural soil-forming processes. In particular, long-term anthropogenic activity in urban areas promotes the significant transformation of the soil cover, which, in extreme cases, leads to the complete destruction of its natural structure and character. These transformations are mainly related to the urbanisation of the municipal environment, war damage, construction and the creation of surface and underground infrastructures, and environmental pollution by various emission sources: municipal, industrial, and communication. They lead to the formation of soils with specific properties, the parameters of which depend on the intensity, direction, and timing of the human impact [38,39,40].

The soils of urban areas, in addition to their function as a carrier of the city’s infrastructure, significantly impact the development of urban greenery and the quality of the city’s water, air, and microclimate. Their degradation, in addition to the negative transformation of other elements of the urban environment, results in the deterioration of the population’s living conditions. Therefore, continuous monitoring of the city’s natural components, including the soil environment, should be pursued. Currently, the study of urban soils is not only of cognitive or scientific importance but is also legally sanctioned [Journal of Laws No. 165, 2002, item 1359]. In most countries of the European Union, knowing the properties of soils is a requirement before soil management can be undertaken, with the ultimate aim of these tasks leading to the development of soil maps for cities. Urban soil mainly occurs in urban green spaces such as parks and gardens and is a repository for contaminants, and it is in such places that people have more direct contact with soil.

According to ecopedological studies carried out in urban parks, *Ch. majus* grows on well and poorly drained soils, despite its anthropogenic nature. As a woodland and desert species, it is tolerant of shade but develops better in relatively good light. One plant (specimen) can produce up to five generative shoots in good light conditions, but no more than one or two in shade and poor drainage conditions [3]. It does not tolerate crusts and compacted soils. In the parks tested, this species grows at the edges of the parks and on open slopes on a loose substrate with good aeration. The soil under *Ch. majus* can be described as slightly acidic (5.6–6.8 in KCl) to alkaline (7.1–7.4 in H_2_O). Only a slight acid reaction in water was found from sites PZiel-3 (6.4) and PSiel-6 (6.3). This type of reaction influences the values of hydrolytic acidity (Hh) and the content of acid cations (Table 3). These soil pH values are determined by the interaction of hydrogen and hydroxide ions in the soil solution and the calcium carbonate content of anthropogenic soils [40]. The paths through the park and the walls around the park are composed of materials containing calcium carbonate, and, during rainfall, the dissolved ions enter the soil. Dust and ash from neighbouring areas also contribute significantly to the alkalinisation of urban soils. The alkaline pH immobilises heavy metals and thus reduces their uptake by plant species.

The content of Corg and Nt in soils under *Ch. majus* communities is very uneven, mainly due to the deposition of external material of a mineral nature in the case of sites located within urban boundaries (PSchoe-2, PSiel-6) or organic sediments from flood waters or agricultural fields. The latter occurs on sites located on less disturbed land outside urban areas (PLes-4). The content of organic carbon and total nitrogen also affects the content of available K, Mg, and available phosphorus. Similar results were found in soils from other urban areas, especially urban green spaces [40,41,42,43]. High total phosphorus (Pt) concentrations ranging from 298 to 940 mg/kg were found at all sites. According to many authors [38,44], phosphorus content of more than 300 mg/kg in the soil indicates that it is caused by various human activities and can therefore be considered an indicator of environmental anthropogenisation. The source of phosphorus in urban parks is related to the fertilisation and maintenance of urban greenery, the accumulation of organic waste, domestic sewage run-off, waste, and domestic animal excrement, which are regular features of urban parks. It can therefore be concluded that total phosphorus accumulation in anthropogenic soils is related to human activities. The high Pt content positively affects the viability of *Ch. majus*, as evidenced by its morphology.

Urban soil, as with natural soil, is one of humanity’s most valuable resources due to its variety of services within the city. Unfortunately, rapid industrialisation and urbanisation during recent decades have dramatically affected the urban soil properties and led to large discharges of pollutants, which inevitably affect the health of the soil, ecosystems, and human populations [45,46]. Emissions in urban areas come from transport (fossil fuel combustion, attrition of parts and tires, petrol and engine oil leaks), coal combustion (power plants and heating), industrial activities (mining, metallurgy, and chemical engineering), building, and waste disposal and incineration, which contaminate the soil and ecosystems [46,47]. Within industrial cities, urban soils (strictly in parks) cover a significant area. They are created by land reclamation, where the parent material is often rubble/post-industrial waste, usually overlain by a 0.5 m layer of fertilised humus material from other sites. Such transported material has different physicochemical properties [39] and determines further soil-forming processes and properties. We analysed the heavy metal content in the near-surface (average 25 cm, rhizome occurrence zone) soil layers under developing *Ch. majus* collected from urban parks (Table 1).

The total content of heavy metals in the soils of recreational and recreational areas is a helpful indicator of the intensity of environmental pollution, used to assess the degree of their degradation [48]. In the superficial soil horizons (humus horizon) in the analysed parks, there was significant variation in the content of individual metals. Among the heavy metals in all the study sites, Pb, Zn, Cd, and Cu (only in APrzem-1) exceeded the permissible limits for land surface pollution for recreation and leisure areas, children’s play areas, arranged parks, and squares [Journal of Laws 2016, item 1395], and the content of trace elements in agricultural soils [48]. Zn exceeded the norm three times (1363.8 mg/kg) in PLes-4, Pb almost three times (526.55 mg/kg) in APrzem-1, and Cd seven times (13.46 mg/kg) in PLes-4 and four times (8.60 mg/kg) in APrzem-1. Spearman’s rank correlation showed that the content of these elements in the soil was closely related.

Numerous studies have confirmed the significant contamination of urban surface soils with Zn, Pb, and Cu [40,41,42,46,47,49,50]. In light of the current legislation in Poland [Journal of Laws 2016, item 1395], the humus horizons of the soils of the studied parks showed permissible content of Mn, Cu (except for APrzem-1), Ni, Cr, Hg, and As. Some other sources of soil contamination involve the use of fertilisers, pesticides, sewage sludge, and organic manures [51]. Plants readily assimilate such elements through the roots. Metallic ions dissolve in water and are retained [52].

Analysing the relationship between the elemental content of the soil and the developing plant species provides reliable information on the transfer of a particular chemical element from the soil to particular plant parts or the lack thereof [53,54]. Every higher plant needs macronutrients and micronutrients for growth, and they are taken up from the soil through the root systems. Most mineral nutrients are bound to the solid phase of the soil and practically inaccessible. They only take them up in a water-soluble form, i.e., from the soil’s liquid phase. This way, heavy metals are also taken up. First, the metals are found in the soil solution (liquid phase), followed by the exchangeable fraction (adsorbed in soil particles). From the metal forms present in the soil solution, simple ions (such as Pb^2+^ or Zn^2+^) are more easily taken up than complex ions [52,53] in natural conditions [55,56]. These reasons may partly account for the increased Zn and Pb content of the analysed *Ch. majus* tissues compared to other elements (Cu, Cr, As, Cd). We observed higher values of Pb (33.92 mg/kg), Cd (3.30 mg/kg), Zn (408.30 mg/kg), Cu (19.53 mg/kg), Ni (3.10 mg/kg), and As (2.60 mg/kg) in the rhizomes, while Mn (293.00 mg/kg), Cr (16.10 mg/kg), and Hg (54 ppb) prevailed in the leaves. Spearman’s rank correlation showed high correlations between the content of these elements (Pb, Zn, Cd, As) (Appendix A) in the soil and rhizomes of *Ch. majus*. A similar regularity was also found by Szentmihályi et al. [28].

The results obtained from some sites (APrzem-1, PZiel-3) correspond with the results obtained by other authors [18] for medicinal plants on the example of Cr (4.35 mg/kg), Zn (57.29 mg/kg), and Cu (15.87 mg/kg) in the aerial parts of plants. Detailed analysis of the mineral composition of the medicinal plant *Ch. majus* in an area unencumbered by human activity was carried out by Sárközi et al. [26]. They analysed the plant and divided it into two parts: the herb and the rhizome. This work divided the analysed material into rhizomes, leaves, and stems. Sárközi et al. [26] obtained similar results to the present study (Table 3) in the case of Cu (18.12 mg/kg—herb, 14.34 mg/kg—rhizome), Mn (22.4 mg/kg—herb, 31.02 mg/kg—rhizome), and Ni (1.42 mg/kg—herb, 2.51 mg/kg—rhizome). In this work, the maximum content of Zn in the rhizome is 408.3 mg/kg, and Sárközi et al. [26] give a value of 141.2 mg/kg, almost three times more. Our results regarding the content of heavy metals (Table 3) are many times higher than those obtained by Ullah et al. [19] and for medicinal plants. The content of Pb, Cd, Zn, Mn, Fe, Cu, and Cr exceeded the limit values for edible and medicinal plants (in the case of Pb, Cd) recommended by the FAO/WHO [57]. Nevertheless, the calculated bioaccumulation factors show that greater celandine is a moderate accumulator of most examined heavy metals.

The soil’s physicochemical properties affect plant species’ viability by making nutrients available. The reflection of plants’ chemical composition results from the richness of the chemical composition of the soil material and the parent rock in nutrients. Thus, there are differences in the concentrations of the elements in the same species in different environments. Relationships between some alkaloids (including isoquinoline) and elements such as Co, Cu, and Zn are genetically determined in *Ch. majus*. This may also impact the content in the aerial part of this species [25,58].

## 5. Conclusions

Although urban parks are anthropogenic ecosystems, they provide habitats and are biodiversity hotspots for many living organisms, including medicinal plants such as *Ch. majus*. As a nitrophilous species, it grows at the edges of parks and on open slopes on loose soil with good aeration and does not tolerate compacted soils. The substrates for *Ch. majus* are predominantly alkaline, rarely slightly acidic. It contains significant amounts of Corg., Nt, and Pt, which influences the soil fertility and indirectly the species growth.
The content of potentially toxic metals in the surface soil layer varied across all sites. The highest content was found for Zn, Pb, and Cd. They exceeded the acceptable standards for recreational areas and cultivated soils.Comparing plant parts, we found the highest concentrations of heavy metals in the rhizomes and leaves, respectively. Zn showed the highest values among the studied elements.Analyses of bioaccumulation (BAF) and translocation factors (TF) showed that *Ch. majus* generally does not accumulate or transport heavy metals from the rhizomes to the plant’s aboveground parts. However, the ability to translocate mercury and chromium into *Ch. majus* leaves means that it should be approached with caution when collecting it from areas with high contamination levels.The relationship among the concentration of heavy metals in the tissues of the studied species and the soil is varied and is related to the chemistry of the initial anthropogenic parent rock for individual research sites.

The determination of elemental content, especially heavy metals, in medicinal plants is important for several reasons: to check the purity of the plant in terms of potentially toxic elements and to establish the presence of these elements in the soil, as the plant takes them up from the soil. Conducting a survey can also serve as a monitor for environmental pollution by repeating the survey in the same area.

## Figures and Tables

**Figure 1 ijerph-20-03887-f001:**
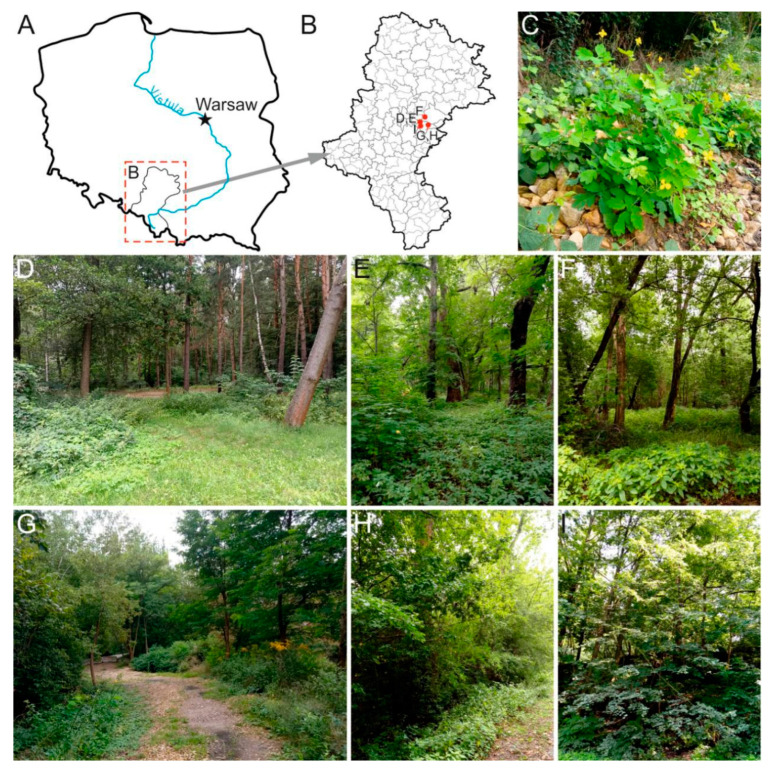
Sampling locations. (**A**) Silesian voivodship on the background of the contour map of Poland; (**B**) sampling sites in the background of the contour map of the Silesian voivodeship; (**C**) photo of *Ch. majus*; (**D**–**I**) photos of sampling sites from Schöen Park (PSchoe-2), allotment gardens near Przemsza river (APrzem-1), Zielona Park (PZiel-3), Leśna Park (PLes-4, PLes-5), and Sielec Park (PSiel-6), respectively.

**Table 1 ijerph-20-03887-t001:** The general features of analysed urban parks.

	Habitat	Type of Soil	Vegetation Types	Ecological Function	Site Coordinates
APrzem-1	Along paths in areas spontaneously overgrown with nitrophilous shrubs	Urbic technosol, mollic technosol, technic regosole	Stand with *Robinia pseudacacia*, *Acer negundo*, and *Sambucus nigra*; at its edge grows *Ch. Majus*, with share of *Impatiens parviflora*, *Arctium tomentosum*, *Urtica dioica*, *Reynoutri japonica*, *Artemisia vulgaris*, and other nitrophilous species	abandoned and derelict	50°17′57.99″ N 19°08′20.76″ E
Pschoe-2	Long park paths on artificial man-made embankments			historic park, culture and entertainment	50°18′03.86″ N 19°08′33.87″ E
Pziel-3	Zone adjacent to hornbeam woodland, habitat is built up by soil and garden rubbish, next to car park			Ecological, sport and recreation	50°20′41.79″ N 19°10′56.63″ E
PLes-4	At the edge of the forest by a closed car park			Ecological, sport and recreation, economic, culture and entertainment	50°18′01.05″ N 19°14′29.87″ E
PLes-5	At the foot of the artificial embankment				50°18′09.76″ N 19°14′43.86″ E
PSiel-6	At the foot of the artificial embankment			Ecological, sport and recreation	50°16′59.08″ N 19°08′34.82″ E

**Table 2 ijerph-20-03887-t002:** Granulometric composition in the analysed sites.

Site	[mm]								
	>10.0	10.0–5.0	5.0–2.0	2.0–1.0	1.0–0.5	0.5–0.25	0.25–0.1	0.1–0.05	<0.05
	[%]								
APrzem-1	5.3	2.5	3.7	6.8	21.1	34.9	18.5	4.1	3.1
PSchoe-2	14.9	8.3	8.5	6.5	13.2	22.9	17.2	4.9	3.6
PZiel-3	-	1.9	2.6	4.1	23.2	39.9	19.3	4.9	4.1
PLes-4	16.2	9.0	8.2	6.7	13.4	23.7	14.7	5.2	2.9
PLes-5	3.4	1.6	2.0	5.1	25.4	34.3	21.7	4.2	2.3
PSiel-6	5.0	2.1	2.7	10.0	20.8	28.9	20.0	5.4	5.1

**Table 3 ijerph-20-03887-t003:** Physico-chemical properties of soils under *Ch. majus*.

Site	pH		Loss on Ignition	Corg.	Nt	C/N	Mgavail.	Kavail.	Pavail.	Pt	H_h_	Al^3+^	H^+^
	H_2_O	KCl											
			[%]				[mg/kg]				[cmol (+)/kg]		
APrzem-1	7.2	6.6	13.36	11.1	0.275	40	158.5	221	52.8	404	1.48	0.10	0.06
PSchoe-2	7.3	6.7	11.83	8.7	0.223	39	139.0	198	39.0	430	1.32	0.02	0.02
PZiel-3	6.4	5.6	9.26	7.2	0.374	19	153.0	178	12.5	298	2.81	0.01	0.12
PLes-4	7.1	6.5	20.27	3.2	0.601	5	322.5	305	146.6	709	1.92	0.02	0.08
PLes-5	7.4	6.8	11.25	10.4	0.280	37	329.5	294	135.9	512	1.41	0.03	0.04
PSiel-6	6.3	5.7	26.56	13.6	0.664	20	308.0	211	143.1	940	4.72	0.04	0.16

**Table 4 ijerph-20-03887-t004:** The content of heavy metals in topsoil under *Ch. majus* community.

Element/Site	APrzem-1	PSchoe-2	PZiel-3	PLes-4	PLes-5	PSiel-6	Permissible Limits *
Pb [mg/kg]	526.55	373.57	119.00	346.23	197.82	247.03	200
Cd [mg/kg]	8.60	7.82	3.13	13.46	7.64	7.81	2
Zn [mg/kg]	1113.90	1038.20	394.50	1363.80	672.00	950.40	500
Mn [mg/kg]	716.00	562.00	344.00	588.00	375.00	484.00	lack
Fe [%]	2.57	2.76	1.44	2.02	1.07	2.10	lack
Cu [mg/kg]	155.79	75.84	21.90	46.06	29.33	50.02	200
Ni [mg/kg]	27.60	33.80	9.40	32.60	12.80	16.50	150
Cr [mg/kg]	25.00	23.80	17.20	17.30	18.30	13.50	200
Hg [ppb]	309.00	330.00	112.00	198.00	128.00	254.00	5
As [mg/kg]	22.40	23.10	8.50	18.70	8.90	18.10	25

* In accordance with the regulations of the Ministry of the Environment of Poland.

**Table 5 ijerph-20-03887-t005:** The content of heavy metals in *Ch. majus* organs.

Element	Plant Tissue	Investigated Site						Limit Value for Edible Plants *
		APrzem-1	PSchoe-2	PZiel-3	PLes-4	PLes-5	PSiel-6	
Pb [mg/kg]	rhizomes	33.92	25.09	8.36	27.47	10.78	11.51	0.43 (edible)
	stems	6.07	4.95	1.85	7.43	6.31	5.20	
	leaves	18.71	7.6	4.99	20.60	9.23	6.17	
Cd [mg/kg]	rhizomes	2.20	3.30	1.73	1.97	1.40	2.70	0.3
	stems	0.67	1.32	0.35	0.77	0.57	0.56	
	leaves	0.56	1.02	0.46	1.07	0.57	0.34	
Zn [mg/kg]	rhizomes	279.50	408.30	235.50	233.70	178.70	284.00	27.4
	stems	150.80	227.50	91.10	134.20	80.60	158.90	
	leaves	112.30	297.40	57.80	190.10	75.60	116.40	
Mn [mg/kg]	rhizomes	181.00	76.00	106.00	79.00	68.00	45.00	2
	stems	72.00	22.00	29.00	29.00	27.00	20.00	
	leaves	293.00	73.00	71.00	76.00	56.00	42.00	
Fe [%]	rhizomes	0.18	0.15	0.08	0.13	0.07	0.11	20 (mg/kg)
	stems	0.06	0.04	0.03	0.05	0.05	0.06	
	leaves	0.17	0.07	0.07	0.14	0.08	0.08	
Cu [mg/kg]	rhizomes	16.98	16.39	11.59	10.38	7.46	19.53	3
	stems	5.57	4.20	3.30	5.04	3.44	5.80	
	leaves	14.24	8.32	7.57	8.22	7.36	7.80	
Ni [mg/kg]	rhizomes	2.80	3.10	1.30	1.80	2.30	1.40	1.63
	stems	0.90	0.70	0.30	0.60	0.80	0.50	
	leaves	1.60	8.40	0.60	1.40	1.30	0.40	
Cr [mg/kg]	rhizomes	3.10	4.00	2.40	2.70	2.90	2.10	0.02
	stems	2.00	1.90	1.50	1.80	2.60	1.70	
	leaves	3.10	16.10	2.20	2.80	4.00	1.90	
Hg [ppb]	rhizomes	37.00	25.00	16.00	16.00	17.00	24.00	Lack inform.
	stems	11.00	11.00	4.00	6.00	10.00	9.00	
	leaves	45.00	45.00	27.00	42.00	54.00	36.00	
As [mg/kg]	rhizomes	2.60	1.10	0.60	2.40	0.70	1.50	
	stems	0.50	0.40	n.d.	0.60	0.60	0.40	
	leaves	1.10	0.50	0.20	1.50	0.50	0.40	

* according to the FAO/WHO Guidance Note.

**Table 6 ijerph-20-03887-t006:** Bioaccumulation factor (BAF) values for heavy metal content in greater celandine tissues.

	Pb	Cd	Zn	Mn	Fe	Cu	Ni	Cr	Hg	As
Leaves	0.04	0.09	0.15	0.19	0.05	0.19	0.09	0.24	0.22	0.04
Stems	0.02	0.09	0.16	0.06	0.03	0.10	0.03	0.10	0.04	0.03
Rhizomes	0.06	0.32	0.33	0.18	0.06	0.29	0.11	0.15	0.11	0.09

**Table 7 ijerph-20-03887-t007:** Heavy metal translocation from rhizomes to stems and leaves of *Ch. majus*.

	Pb	Cd	Zn	Mn	Fe	Cu	Ni	Cr	Hg	As
Leaves	0.57	0.30	0.52	1.10	0.85	0.65	1.08	1.75	1.84	0.47
Stems	0.27	0.32	0.52	0.36	0.38	0.33	0.30	0.67	0.38	0.28

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
