# Peer review of "Relationships between Heavy Metal Concentrations in Greater Celandine (*Chelidonium majus* L.) Tissues and Soil in Urban Parks"

_ijerph, 2023, doi:10.3390/ijerph20053887_

Round 1
Reviewer 1 Report
The paper addresses the relationship between the content of potentially toxic metals in the plant-soil system in urban areas. Since the habitats of Ch. majus are anthropogenically transformed areas, the authors attempted to determine the chemical composition of the tissues of the above species, which is a medicinal plant. Works on the elemental composition of Ch. majus are scarce, and the present work brings new results in this area. Knowledge of the chemical composition of medicinal plants is essential. The work is correctly done, and the results are generally presented clearly. Below a few comments:
L: 18: check the scientific style of “H2O”.
L: 32: Do not start any sentence with abbreviation. This applies to whole manuscript.
“In nature, Ch. majus grows in Asia, Europe, and the Mediterranean”. Any reference?
What are folk medicines?
“The use of the medicinal plant in different indication fields is well known”. Please rewrite this sentence.
L: 92: “chemical composition of the soil”, does not seem suitable words for soil.
unify: total N to Nt,
line 177 - humus level change to humus horizon (across the text, if any).
Heading 2.2: change “descriptions” to “traits/characteristics”.
In what plant community does it grow outside the parks? Also, in transformed sites?
L: 167: Better to mention the fill names+symbols at their first mention.
The results section seems better and ready to go.
L: 345-349: the sentence is too big and not understandable, rewrite.
Are all these parks created on transported material (external parent rock) or on the natural ground? This information should be included in subsection 2.1. Such information essential considering the content of heavy metal in soil.
Conclusion should be based on the actual results obtained and future recommendations.
Author Response
In an attachment

Reviewer 2 Report
Dear authors, I need to clarify some points.
Which soil classification standards are you using? I am not familiar with the term Humus horizon, if this horizon has been introduced in a specific soil classification manual, please add the reference. Personally, I am familiar only with the USDA soil classification (O, A, B, etc). Please clarify.
Line 38: add botanical to family "this botanical family".
Must improve the quality of the tables, similar to Table 6 or 7 which have a good presentation.
why most of your analysis was significant at p=0.05? is this really significant? I only noticed Ni with a p-value = 0.01.
If heavy metals are not transported, generally, how do you explain the highest concentrations of heavy metals in leaves? I would recommend checking and reorganizing your conclusions.
Author Response
In an attachment

Reviewer 3 Report
The manuscript contains some good piece of work which deals with the influence of heavy metal concentrations in plant tissues and urban soil.. The paper exhaustively deals with all the parameters and is written well. At some points the paper needs some more improvement..
In the abstract pl line no 20- write first nitrogen content and phosphate content and then abbreviate it as Nt and Pt.
Literature review on effects of heavy metals on soils needs to be given pl refer the following papers and discuss the physiochemical changes heavy metals bring in natural soils.
10.1007/s40098-022-00643-x, DOI: 10.1007/s11356-022-19551-x, DOI: 10.1061/JHTRBP.HZENG-1200, http://hdl.handle.net/10603/421178
Line 208, pl classify the type of soil based on the data obtained such as kaolinitc, illitic, montmorillonitic or peat etc.use USGCS system of classification.
Line 229 anthropogenic origin….. based on mineral content but in the selection of sampling points the authors should describe the area giving any such source like the vicinity of pollutant source like a factory or any other such unit else it amounts to just speculation, also this should become a reason for selecting such a sampling point.
Line 147 pl first give the main objective of performing statistical analysis for this type of work and its significance.
Fig 3 line 185 the concentration of heavy metals is higher in Bio- MPs than PE-MPs and control soil similarly in fig 4 pl check and there seems to be confusion whether it is concentration or probability distribution.
Table 4 compare these values with the standard permissible limits only then the readers would be able to judge the gravity of the situation. Nobody will remember standards hence it has to be reiterated and compared. One extra column in table 4 shall suffice for standards
Discussion part seems to be overwhelming and needs to be pruned and made more crisp and analysis should be based on data determined.
The conclusion please list the important outcomes of this work point wise and give one final takeaway to readers from this work.
Author Response
In an attachment

Reviewer 4 Report
medicinal species Chelidonium majus in 6 urban parks……Where in the world?
What is “greater celandine”
clumps of Ch. majus………Will it not be C. majus?
Homogenise in the text about usage of term “Ch. majus L., greater celandine,”
In this line………….The use of the medicinal plant in different indication fields is well known.
May be cited
Multi trait Pseudomonas sp. isolated from the rhizosphere of Bergenia ciliata acts as a growth promoting bio-inoculant for plants
Frontiers in Sustainable Food Systems 7, 2
“Evaluation of aflatoxin contamination in crude medicinal plants used for the preparation of herbal medicine, Oriental Pharmacy and Experimental Medicine 19 (2), 137-143”
Not a very attractive line as per theme” In this way, we can 101 find an answer to the question of whether such a medicinal plant can be used in medicine.”
In table 2 H2O…..dot is there.plz remove
Why this particular area was selected as per medicinal plant analysis.
How this work is very important to herbal medicine industry must be written to actually showcase the work.
How this heavy metal accumulation may be avoided?
This paper is related to phytoremediation…see if this is useful
Edit this ----------Table 1. This is a table. Tables should be placed in the main text near to the first time they are cited. 1
Make a new conclusive figure
Author Response
In an attachment
